Neglected Tropical Diseases

# Epidemiology of intestinal parasite infections and multiparasitism and their impact on growth and hemoglobin levels during childhood in tropical Ecuador: A longitudinal study using molecular detection methods

**Rojelio Mejia[1], Irina Chis Ster[2], Martha E. Chico[3], Irene Guadalupe[3], Andrea Arévalo-Cortés[3], Andrea Lopez[4], Aida Y. Oviedo-Vera[3], Philip J. Cooper** [2,3,4]*

**1** National School of Tropical Medicine, Baylor College of Medicine, Houston, Texas, United States of America, **2** Institute of Infection and immunity, St George's University of London, United Kingdom, **3** Fundación Ecuatoriana Para Investigación en Salud, Quito, Ecuador, **4** School of Medicine, Universidad Internacional del Ecuador, Quito, Ecuador

\* pcooper@sgul.ac.uk

## Abstract

### Background

There are few longitudinal epidemiological studies of intestinal parasitic infections (IPI) and their health effects. We studied the epidemiology and determinants of IPI and multiparasitism during childhood using molecular methods for parasite detection and analysed their effects on growth and hemoglobin levels.

### Methods

Random sample of 401 children from an Ecuadorian birth cohort followed up to 8 years of age. Data on environmental and sociodemographic characteristics were collected by questionnaires. Stool samples were collected, and weight, height, and hemoglobin levels were measured at 7 and 13 months, and 2, 3, 5, and 8 years. Stool samples were analysed using multi-parallel quantitative polymerase chain reaction for the presence of soil-transmitted helminth (STH) (*Ascaris lumbricoides*, *Trichuris trichiura*, *Ancylostoma* spp. *Necator americanus*, and *Strongyloides stercoralis*) and protozoal (*Giardia lamblia*, *Entamoeba histolytica*, and *Cryptosporidium* spp.) parasites. Associations between risk factors and infections, and between infections and nutritional outcomes were estimated using generalized estimating equations applied to longitudinal binary or continuous outcomes.

### Results

IPI were observed in 91.3% of the cohort during follow-up with peak proportions between 3 and 8 years, while multiparasitism increased more gradually (32.5% at

**Data availability statement:** All relevant data are within the paper and its Supporting Information files.

**Funding:** Funding for this study was provided in part by the Wellcome Trust, London, United Kingdom (WT 088862/Z/09/Z); Baylor College of Medicine Center for Globalization Pilot Project and an Early Career Thrasher Research Fund award (#12011); and Universidad Internacional del Ecuador, Quito, Ecuador (Grant No. UIDE-EDM01-2015). The funders had no role in study design, data collection and analysis, decision to publish, or preparation of the manuscript.

**Competing interests:** The authors have declared that no competing interests exist.

8 years). Factors significantly associated with multiparasitism included lower birth order, day care, Afro-Ecuadorian ethnicity, urban residence, lower household income, and maternal STH infections. IPI during follow-up were associated with lower hemoglobin (difference = -0.102, 95% CI -0.192 - -0.013, P = 0.025), height-for-age (difference = -0.126, 95% CI -0.233 - -0.019, P = 0.021) and weight-for-age (difference, -0.129, 95% CI -0.257 - -0.022, P = 0.018) z scores. Multiparasitism had the strongest negative effects on growth (height-for-age, -0.289, 95% CI -0.441- -0.137, P < 0.001; weight-for-age, -0.228, 95% CI -0.379 - -0.077, P = 0.003), with some evidence of greater effects with greater number of parasite species.

## Conclusion

IPI infections and multiparasitism were frequent during early childhood in this Ecuadorian cohort. IPI was associated with reduced weight, height, and hemoglobin trajectories while children with multiparasitism had the greatest growth deficits. Our data highlight the adverse health effects of multiparasitism during childhood in endemic settings and the need for integrated programmes of control and prevention to eliminate associated morbidity.

## Author summary

Intestinal parasite infections (IPI) and co-infections with several parasite species (multiparasitism) are considered to cause significant disease among young children in low and middle-income countries (LMICs), and elimination of mortality and morbidity attributable to IPI among preschool children by 2030 has been prioritized by WHO. We followed a random sample of children from a birth cohort to 8 years of age in a rural area of coastal Ecuador for the acquisition of IPI with parasite detection being done using highly sensitive molecular methods. Almost all children acquired IPI during follow-up and approximately half had multiparasitism by 8 years. Risk of multiparasitism was associated with indicators of marginalisation (Afro-Ecuadorian ethnicity and peri-urban residence), household poverty (low household income), exposure to other children inside (being lower in the birth order) and outside (day care attendance) the household, and parasite infections among household members. IPI during childhood was associated with reduced growth (height and weight), and lower hemoglobin levels, while children with multiparasitism suffered the greatest growth deficits. There remains a need for the implementation of integrated control strategies including access to clean water and improved sanitation, health education, and preventive chemotherapy that could target whole communities to reduce the infection reservoir. Preventive chemotherapy with broad-spectrum drugs (e.g., nitazoxanide) with anthelmintic and antiprotozoal effects, even if targeted only to pre-school and school-age children and women of child-bearing age, could improve growth during childhood in marginalized populations living in endemic settings.

## Introduction

Intestinal parasite infections (IPI) including soil-transmitted helminths (*Ascaris lumbricoides*, *Trichuris trichiura*, and hook-worms [*Necator americanus* and *Ancylostoma duodenale*] and protozoa (primarily *Giardia lamblia*, *Entamoeba histolytica*, and *Cryptosporidium* spp.), have a worldwide distribution, and represent a significant public health challenge in many low and middle-income countries (LMICs), particularly those with tropical and subtropical climates [1,2]. IPI are transmitted through contaminated soil, water, and food in regions with inadequate access to sanitation and potable water [3]. These parasitic infections are associated with malnutrition, anemia, impaired cognitive and physical development, and decreased productivity in affected populations [4]. Children and marginalized communities are especially vulnerable to infection, perpetuating a cycle of poverty and disease [2].

Intestinal parasitic infections were estimated to infect almost 2 billion humans in 2019 [5]. Approximately 1 billion were estimated to be infected with STH parasites causing 1.9 million DALYs [6], and up to 300 millions suffer severe morbidity [7]. Concurrent infections with more than 1 parasite species are commonly observed [8,9], and individuals with these multiparasitic infections tend to harbor more intense infections [10,11]. Even low intensity of multiparasitic infections may result in clinically significant morbidity [12,13].

WHO has prioritized the elimination of mortality and morbidity related to IPI among preschool children by 2030 [14]. Although the Ecuadorian government has made efforts to combat STH through mass drug administration (MDA) programs, the disease burden remains high, particularly among rural marginalized communities [15,16], and there are no programs for the control of endemic enteric protozoal pathogens. A better understanding of the epidemiology of IPI and multiparasitism in Ecuador will be important for the design of better-targeted control strategies, as well as assessing the impact of current control efforts. More sensitive molecular methods such as quantitative PCR, a platform that has been previously validated in this and other settings [9,17], provide useful tools with which to improve parasite detection and evaluate the impact of these infections on childhood growth and nutrition.

There are few studies of the epidemiology of IPI and multiparasitism from Latin America that consider infections with STH and protozoal pathogens using highly sensitive molecular methods for detection, and which evaluate the potential impact of IPI and multiparasitism on early childhood growth and anemia. In the present study, we used molecular methods to analyze longitudinally collected stool samples from a sub-sample of a birth cohort to detect IPI to understand better their epidemiology and risk factors in this setting during the first 8 years of life and evaluate their impact on growth and nutrition.

## Methods

### Ethics statement

The study protocol was approved by the ethics committees of the Hospital Pedro Vicente Maldonado (2005), Universidad San Francisco de Quito (2010), and Baylor College of Medicine, H-33219 (2013). Informed written consent was obtained from the child's mother for collection of stool samples and data. Individuals with positive stools for soil-transmitted helminth infections were treated with a single dose of albendazole if aged 2 years or greater and with pyrantel pamoate if aged <2 years, according to Ecuadorian Ministry of Public Health recommendations [18]. Anti-protozoal treatments (metronidazole or tinidazole) were offered to symptomatic children with *E. histolytica* or *G. lamblia* trophozoites in fresh stool samples.

### Study area and population

Detailed methodology of the study objectives, design, follow-up and sample and data collection for the ECUAVIDA cohort study is provided elsewhere [19]. Briefly, the ECUAVIDA cohort was a population-based birth cohort of 2,404 newborns whose families lived in the rural district of Quinindé, Esmeraldas Province, and were recruited around the time of birth at

the Hospital Padre Alberto Buffoni in the town of Quinindé between November 2005 and December 2009. This population-based cohort was designed to study the effects of early life infections on the development of allergy and allergic diseases in childhood. The present analysis focused on a random sample of 401 children with stool samples collected at 13 months of age. The district of Quinindé is largely agricultural where the main economic activities relate to the cultivation of African palm oil and cocoa. The climate is humid tropical with temperatures generally ranging 23-32°C with yearly rainfall of around 2000-3000mm. Inclusion criteria into the cohort were being a healthy baby, collection of a maternal stool sample, and planned family residence in the district for at least 3 years.

### Study design and sample collection

Children were followed-up from birth to 8 years of age with data and stool samples collected at 3, 7, and 13 months, and 2, 3, 5, and 8 years of age. Follow-ups were done either by scheduled visits to a dedicated clinic at HPAB or by home visits. At the initial home visit, a questionnaire was administered to the child's mother by a trained member of the study team to collect data on potential risk factors [19]. Maternal questionnaires were repeated when the child was 7 and 13 months and 2, 3, 5, and 8 years of age.

### Anthropometric and nutritional measurements

Anthropometric measurements were done as described [20]. Briefly, first measurements of weight and height were done between birth and 2 weeks of age and then repeated periodically during clinic and home visits at 7, 13, 24, 36, 60, and 96 months. At each observation time, length/height (cm) and weight (kg) were measured in duplicate by trained members of the research team. Children were weighed without clothes or with light underwear on portable electronic balances (Seca, Germany) accurate to within 100 grams. Length/height measurements were done using wooden infantometers/stadiometers to within 0.1 cm. Z scores for weight-for-age (WAZ), height-for-age (HAZ) and body mass index-for-age (BAZ) at each observation time were calculated using WHO growth standards [21]. Hemoglobin levels (g/dL) were derived from haematocrits obtained following the centrifugation of capillary blood using standard laboratory procedures. Anemia was defined as less than 11 g/dL.

### Stool examinations

Stool samples were collected from household members around the time of birth of the cohort child. Samples were examined using four microscopic techniques to detect and/or quantify STH eggs and larvae including direct saline wet mounts, formol-ether concentration, modified Kato-Katz, and carbon coproculture [22]. All stool samples were examined using all 4 microscopic methods where stool quantity was adequate. A positive sample for STH was defined by the presence of at least one egg or larva from any of the above detection methods.

### Molecular analyses for intestinal parasite infections

An aliquot of stool was preserved in 90% ethanol at -30°C for molecular analyses. Samples were processed using FastDNA SPIN Kit for Soil (MP Biomedicals, Santa Ana, California, USA). Stool DNA was analysed by multi-parallel quantitative real-time polymerase chain reaction (qPCR) to detect *Ascaris lumbricoides*, *Trichuris trichiura*, *Ancylostoma* spp., *Necator americanus*, *Strongyloides stercoralis*, *Giardia lamblia*, *Entamoeba histolytica, and Cryptosporidium* spp. [9]. All reactions were performed in duplicate and a total volume of 7μL containing 3.5μL of TaqMan fast mix (Applied Biosystems, Waltham, MA), 2μL of template DNA, and 1.5μL of primers (final concentration of 900nM) and FAM-labelled minor groove binder probe (final concentration of 250nM). All extractions included a positive control (diluted plasmid standards), a negative control (no template), and 1 μl of internal amplification control (IAC) plasmid at a concentration of 100 pg/μl, containing a unique 198-bp sequence and detected by qPCR using PCR primers and probe sequences as described [23]. Positive qPCR were considered on concordant duplicate samples with cycle thresholds (Ct) under 38 as previously [9].

## Statistical analysis

Potential risk factors considered included individual (for participant child - sex, age, birth delivery, birth order, duration of breast feeding, day care attendance during first 3 years of life, and anthelmintic treatments), maternal (age, educational status, and ethnicity), and household (socioeconomic status, area of residence, overcrowding, monthly income, construction materials, material goods [fridge, TV, radio, and HiFi], type of bathroom, pets, agricultural activities, and STH infections in household members). Socioeconomic variables were combined to create a socio-economic status (SES) index by using principal components analysis for categorical data as described [22]. The presence or absence of any IPI, any STH, and any intestinal protozoal infection from 3 months to 8 years defined the main longitudinal binary outcomes. Stool sample collections were done periodically as described with some time variation although for analyses purposes these were generically considered as 7, 13, 24, 36, 60 and 96 months of age. All stool samples were analyzed using qPCR irrespective of the results of microscopy, and the only results of qPCR analyses were used in this analysis. We used generalized estimation equations models (GEE) to fit population-averaged models [24] to understand the effects of age, childhood, parental and household characteristics, and household STH infections (any member, parents, and siblings) on the age-dependent risk of acquiring IPI (or any STH or any protozoa) during childhood. Binary random effects models were also considered [24,25]. We used the assumption of an unstructured correlation structure [24–26] and, for the sake of simplicity and given that our questions were addressed at the population level, we have presented and commented on population average estimates only. Associations and their uncertainties were measured by odds ratios (OR) and their corresponding 95% confidence intervals (CIs). ORs derived from these longitudinal models estimated associations between potential explanatory variables and the age-dependent risk of parasite infection outcomes. Minimally adjusted models (for age and its nonlinearities reflected by terms of higher powers) investigated the sole association of each factor on IPI outcomes. Multivariable models were subsequently built using variables with P<0.1 in minimally adjusted models and the smallest associated quasi-likelihood value under the independence model criterion (QIC) for GEEs [26,27] on a complete data sample. The final most parsimonious model using a complete data sample was then fitted back to the original data on as many observations as possible. Because longitudinal cohorts are subject to attrition at follow-up, we analysed patterns in missing data for IPIs and did sensitivity analyses. GEE estimations were based on the missing completely at random assumption [26,28]. Random effects, based on maximum likelihood estimation, were also fit under missing at random assumption [26,28] and tended not produce very different estimates in terms of ORs or their standard errors [26,27]. Predictions for age-dependent risk of any IPI, any STH, and any intestinal protozoal infection were displayed against raw data. All estimates were age-adjusted and accounted for the hierarchical structure of outcomes. We used longitudinal panel data analysis [26] to explore the longitudinal effects of any IPI, multiparasitism, any STH, and any intestinal protozoal infection on parameters of growth (using z-scores for WAZ and HAZ as continuous longitudinal outcomes), nutrition (hemoglobin [continuous] and anemia [binary]), and presence of eosinophilia (binary, defined as>= 500 cells/µL) during childhood. The assumption is that the time-varying covariates remain constant during a panel time span. The GEE and mixed modelling estimation yielded similar findings. Predictions for z-scores overall and by the variables which displayed most marked differences were plotted against age. Statistical significance was inferred by P<0.05. All statistical analyses were done using Stata version 17 (StataCorp, College Station, TX, 2021).

## Results

### Sample analysis

We analysed 2,005 stool samples collected periodically from 401 children between 3 months and 8 years of age (3 [211], 7 [267], 13 [400], 24 [274], 36 [218], 60 [321], and 96 [314] months). Median number of stool samples analysed per child was 4 (Q1-Q3, 3-5).

## Age-dependent risk of IPI and multiparasitism

The proportion of samples collected that were positive for each parasite during the first 8 years of life are shown in Fig 1A for STH and Fig 1B for intestinal protozoa. During follow-up, 91.3% of children had an infection with any IPI documented at least once with equivalent frequencies for any STH and any protozoa being 51.6% and 87.3%, respectively (S1 Table). Peak risk and age of peak for the different parasites were: *A. lumbricoides* (16.9% at 8 years), *T. trichiura* (24.5% at 8 years), *Ancylostoma* spp. (4.5% at 8 years), *N. americanus* (0.9% at 3 years), *S. stercoralis* (5.1% at 3 years), *G. lamblia* (65.6% at 8 years), *Cryptosporidium* spp. (5.9% at 2 years), and *E. histolytica* (7.4% at 8 years). Age-dependent risk of infection with any IPI, any STH infection, and any protozoal infection are shown in Fig 2A–2C. Similar patterns were observed for the 3 outcomes with rapid acquisition of infections during the first 2 years of life after which infection risk reached a plateau with a slight decrease at 5 years. Peak risk of any STH, any protozoa, and any IPI was 34.4%, 68.0%, and 69.5%, respectively, at 8 years of age. Age-dependent risk of infection with multiple parasites (>=2 different) species is shown in Fig 2D and indicated that multiparasitic infections were acquired more slowly with increasing age reaching 32.5% by 8 years.

## Factors associated with age-dependent risk of IPI and multiparasitism

Age-dependent risk for any IPI, STH, and intestinal protozoal infections predicted by the longitudinal models against observed proportions are shown in Fig 2. Predictions were close to observed values – observed values were included in the 95% CIs of the predictions except for any STH and IPI at 7 months. Descriptive statistics for distributions of individual, parental, and household characteristics by any IPI, any STH, and any protozoal infections are provided in S1 Table. Age-adjusted and multivariable associations between these factors and average risk of any IPI during childhood are shown in Table 1. Age-adjusted and multivariable associations for any STH or any protozoal infections are shown in

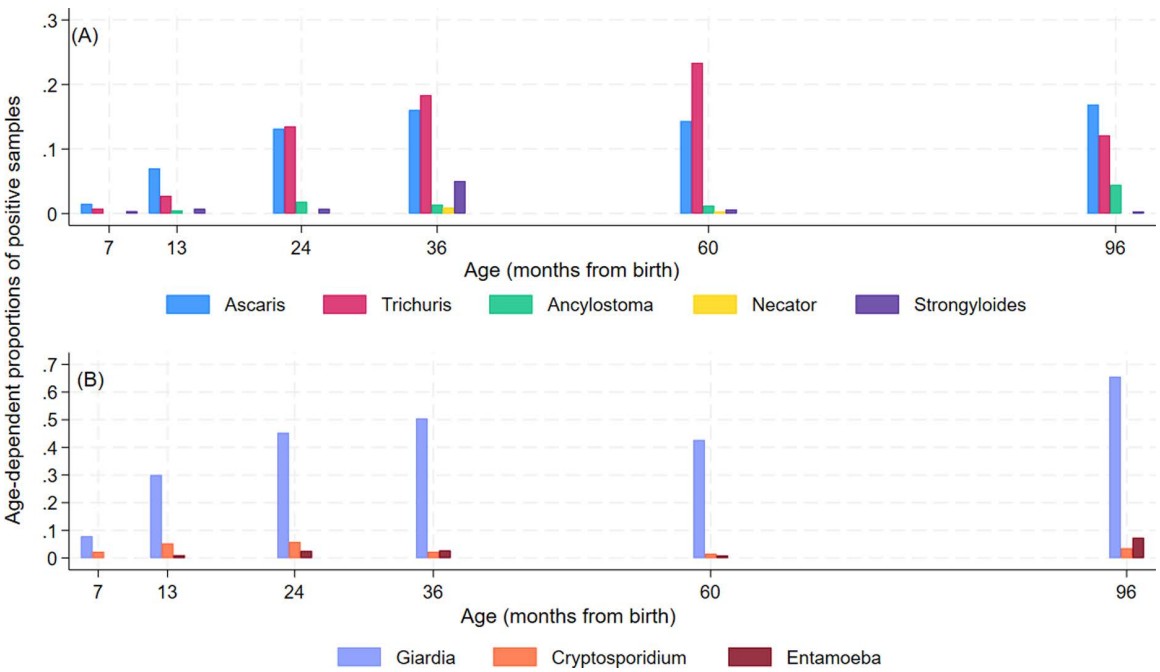

**Fig 1. Age-dependent proportions of positive samples for soil-transmitted helminth (STH) (A) and protozoal (B) parasites from 7 months to 8 years of age in 401 children from a birth cohort.** STH were *Ascaris lumbricoides*, *Trichuris trichiura*, *Ancylostoma* spp., *Necator americanus*, and *Strongyloides stercoralis*. Protozoal parasites were *Giardia lamblia*, *Cryptosporidium* spp., and *Entamoeba histolytica*.

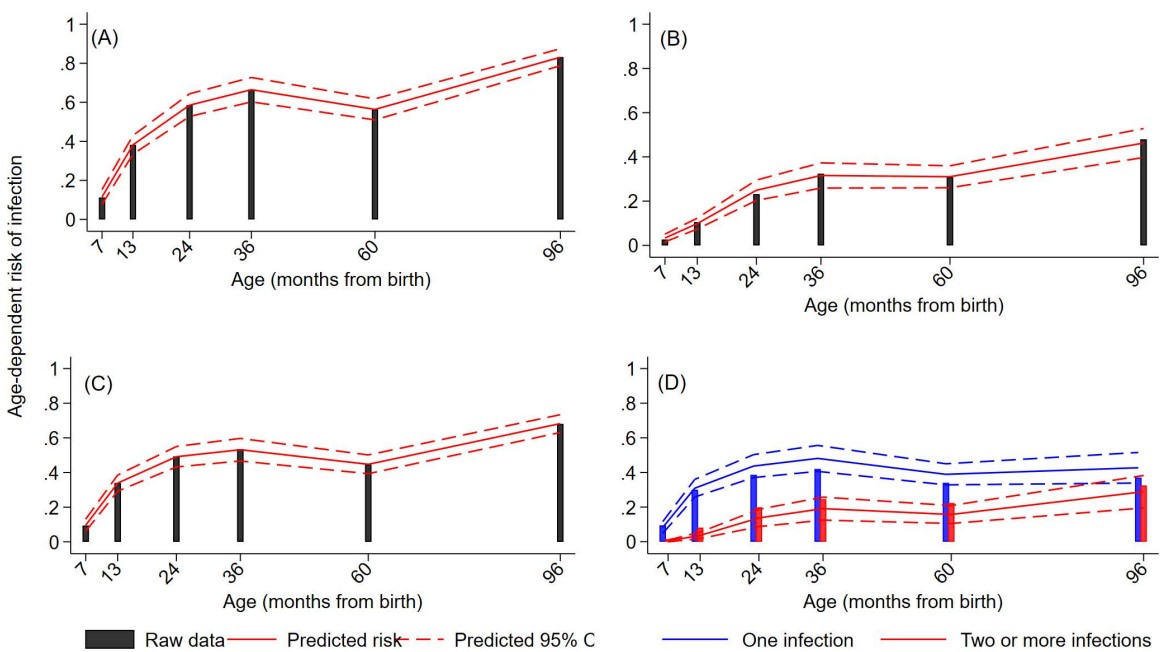

**Fig 2. Age-dependent predicted risk of infection with intestinal parasites from 7 months to 8 years of age in 401 children from a birth cohort.** A – Any intestinal parasite (Parasite) infection. B- any soil-transmitted helminth (STH) parasite. C – any intestinal protozoal (Protozoa) parasite. D – Multiparasitism. Estimated curves and 95% Confidence intervals (dotted lines) are shown against the raw data.

S2 and S3 Tables, respectively. In multivariable analyses, factors significantly associated with greater risk of IPI during childhood included being lower in the birth order (≥5th vs. 1st-2nd), maternal Afro-Ecuadorian ethnicity, and maternal STH infections. Different patterns of risk factors were associated with any STH and any protozoal infection in multivariable models (S1 and S2 Tables): any STH was associated with maternal Afro-Ecuadorian ethnicity, household overcrowding and agricultural exposure, while any protozoa were associated with being lower in the birth order, daycare and urban residence. Both outcomes were positively associated with caesarean birth delivery and maternal STH infections. Models that adjusted for any STH or any protozoa in the respective multivariable models with any protozoa and any STH as outcomes showed significant associations for the corresponding exposure (effect of any STH on any protozoa, adj. OR 2.39, 95% CI 1.80-3.17, P<0.001; effect of any protozoa on any STH, adj. OR 2.65, 95% CI 1.88-3.75, P<0.001). We examined factors associated with multiparasitism stratified as an ordinal multivariate outcome (i.e., 0, 1, 2, and ≥3 IPI). Age-adjusted and multivariable analyses are shown in Table 2. In these models, a single OR represents all comparisons for 3 vs. 0-2, 2-3 vs. 0-1, and 3-1 vs. 0 IPI. In multivariable analyses, factors significantly associated with an increasing number of IPIs in the same individual were being lower in the birth order (>=5th vs. 1st-2nd), day care, maternal Afro-Ecuadorian ethnicity, urban residence, lower household income, and maternal STH infections. Any protozoal infection was strongly associated with any STH infection (as outcome) in a multivariable model (adj. OR 2.65, 95% CI 1.88-3.75, P<0.001) and vice versa (adj. OR 2.39, 95% CI 1.80-3.17, P<0.001).

**Effects of IPI and multiparasite infections on childhood growth trajectories**

Age-adjusted and multivariable associations between infections with any IPI, any STH, and any protozoa and growth trajectories are shown in Tables 3 and S4, respectively. In multivariable analyses, the group of children infected with any IPI or any STH had significantly lower height-for-age (HAZ) and weight-for-age (WAZ) z core trajectories during the first

**Table 1. Age-adjusted and multivariable estimates of associations for child, maternal, and household factors with any intestinal parasite infection from 7 months to 8 years of age.**

| VARIABLES | CATEGORIES | Age-adjusted | | | | Multivariable | | | |
|---|---|---|---|---|---|---|---|---|---|
| **CHILDHOOD FACTORS** | | OR | P-value | 95% CI low | 95% CI high | OR | P-value | 95% CI low | 95% CI high |
| **Age** | | **1.557** | **<0.001** | **1.323** | **1.833** | **1.007** | **<0.001** | **1.006** | **1.008** |
| **Age²** | | **0.987** | **<0.001** | **0.980** | **0.994** | **0.9993** | **<0.001** | **0.9992** | **0.9994** |
| **Age³** | | **1.0002** | **0.006** | **1.0001** | **1.0003** | **1.00002** | **<0.001** | **1.00001** | **1.00003** |
| **Sex** | **Female vs. Male** | 0.830 | 0.083 | 0.673 | 1.024 | | | | |
| **Birth order** | **3rd-4th vs. 1st-2nd** | **1.328** | **0.020** | **1.045** | **1.687** | 1.121 | 0.122 | 0.950 | 1.544 |
| | **≥5th vs. 1st-2nd** | **1.659** | **0.001** | **1.236** | **2.228** | **1.525** | **0.005** | **1.133** | **2.053** |
| **Delivery mode** | **Vaginal vs. Caesarean** | **0.703** | **0.005** | **0.551** | **0.897** | | | | |
| **Exclusive breastfeeding** | **Per month** | 1.026 | 0.227 | 0.984 | 1.069 | | | | |
| **Day care to 3 years** | **Yes vs. No** | **2.148** | **<0.001** | **1.586** | **2.910** | | | | |
| **Anthelmintic treatment (TV)** | **Yes vs. No** | 0.918 | 0.560 | 0.690 | 1.223 | | | | |
| **MATERNAL FACTORS** | | | | | | | | | |
| **Age (years)** | **21-29 vs. ≤ 20** | 0.893 | 0.385 | 0.692 | 1.152 | | | | |
| | **≥30 vs. ≤ 20** | **0.694** | **0.015** | **0.517** | **0.931** | | | | |
| **Ethnicity** | **Non-Afro vs. Afro** | 0.787 | 0.248 | 0.525 | 1.181 | **0.617** | **<0.001** | **0.485** | **0.786** |
| **Educational status** | **Primary vs. illiterate** | 0.827 | 0.257 | 0.596 | 1.148 | | | | |
| | **Secondary vs. illiterate** | **0.639** | **0.015** | **0.446** | **0.917** | | | | |
| **HOUSEHOLD FACTORS** | | | | | | | | | |
| **Socio-economic status** | **Medium vs. Low** | **0.760** | **0.033** | **0.590** | **0.978** | | | | |
| | **High vs. low** | 0.838 | 0.178 | 0.648 | 1.084 | | | | |
| **Area of residence** | **Rural vs. urban** | **0.706** | **0.002** | **0.565** | **0.882** | | | | |
| **Overcrowding** | **≥3 vs. < 3** | **1.402** | **0.002** | **1.132** | **1.736** | | | | |
| **Monthly income (basic basket)** | **≥1 vs. < 1** | **0.509** | **0.013** | **0.299** | **0.867** | | | | |
| **House construction** | **Cement/brick vs. wood/bamboo** | 1.110 | 0.371 | 0.883 | 1.395 | | | | |
| **Dog in house (birth)** | **Yes vs. No** | 1.084 | 0.621 | 0.787 | 1.492 | | | | |
| **Cat in house (birth)** | **Yes vs. No** | 1.169 | 0.328 | 0.855 | 1.600 | | | | |
| **Agriculture (TV)** | **Yes vs. No** | 0.915 | 0.407 | 0.742 | 1.129 | | | | |
| **Bathroom (TV)** | **WC vs. latrine** | 0.971 | 0.818 | 0.753 | 1.251 | | | | |
| **HOUSEHOLD STH INFECTIONS** | | | | | | | | | |
| **Mother** | **Yes vs. No** | **2.023** | **<0.001** | **1.629** | **2.513** | **1.828** | **<0.001** | **1.469** | **2.275** |
| **Father** | **Yes vs. No** | **1.510** | **0.022** | **1.063** | **2.146** | | | | |
| **Siblings** | **Yes vs. No** | **1.509** | **0.008** | **1.114** | **2.045** | | | | |
| **Any in household** | **Yes vs. No** | **1.498** | **0.001** | **1.185** | **1.895** | | | | |

Odds ratios (ORs) and 95% confidence intervals (CI) were estimated by fitting (age, age², age³ and age⁴ -not shown) - adjusted longitudinal models using generalized estimating equations. Longitudinal binary outcomes were defined by presence/absence of any IPI detected in stool samples from children during follow-up. Population-average models were fit under missing completely at random assumption for unobserved data points. Characteristics are at time of birth of child (birth) or time-varying (TV) over the course of follow-up. STH—soil-transmitted helminth infections. Afro- Afro-Ecuadorian. Anthelmintic treatments—maternal reports of at least one anthelmintic treatment during the previous year. Monthly household income was classified according to receipt of an income sufficient to meet the basic needs of 4 persons (or 1 family basket) or US$480 in 2008. Household overcrowding was defined as 3 or more people per sleeping room. Agricultural exposures were defined by living on a farm or having at least weekly visits to a farm. Any infected with STH in the household represented any member of the child's household with a positive stool sample collected around the time of birth of the cohort child.

**Table 2. Age-adjusted and multivariable estimates of associations for child, maternal, and household factors with multiparasitism defined as an ordinal longitudinal outcome (0, 1 and ≥2 infections) from 7 months to 8 years of age.**

| Variables | Categories | Age-adjusted | | | | Multivariable | | | |
|---|---|---|---|---|---|---|---|---|---|
| | | OR | P-value | 95%CI low | 95%CI high | OR | P-value | 95% CI low | 95% CI high |
| **CHILDHOOD FACTORS** | | | | | | | | | |
| **Age** | | **2.6863** | **<0.001** | **1.6299** | **4.4275** | **2.8813** | **<0.001** | **1.7088** | **4.8584** |
| **Age$^2$** | | **0.9507** | **0.005** | **0.9175** | **0.9850** | **0.9457** | **0.003** | **0.9113** | **0.9815** |
| **Age$^3$** | | **1.0013** | **0.023** | **1.0002** | **1.0024** | **1.0014** | **0.013** | **1.0003** | **1.0026** |
| **Sex** | **Female vs. Male** | 0.768 | 0.089 | 0.566 | 1.041 | | | | |
| **Birth order** | 3$^{rd}$-4$^{th}$ vs. 1$^{st}$-2$^{nd}$ | 0.895 | 0.660 | 0.546 | 1.467 | 1.365 | 0.059 | 0.989 | 1.885 |
| | ≥5$^{th}$ vs. 1$^{st}$-2$^{nd}$ | **2.231** | **0.006** | **1.265** | **3.933** | **1.768** | **0.006** | **1.196** | **2.612** |
| | (3$^{rd}$-4$^{th}$ vs. 1$^{st}$-2$^{nd}$)×Age | **1.015** | **0.001** | **1.006** | **1.024** | | | | |
| | (≥5$^{th}$ vs. 1$^{st}$-2$^{nd}$) ×Age | **0.999** | **0.816** | **0.989** | **1.009** | | | | |
| **Delivery mode** | **Vaginal vs. Caesarean** | **0.637** | **0.012** | **0.448** | **0.906** | | | | |
| **Exclusive breastfeeding** | **Per month** | 1.040 | 0.194 | 0.980 | 1.104 | | | | |
| **Day care to 3 years** | **Yes vs. No** | **2.296** | **<0.001** | **1.538** | **3.426** | **1.503** | **0.042** | **1.015** | **2.227** |
| **Anthelmintic treatment (TV)** | **Yes vs. No** | 0.857 | 0.328 | 0.630 | 1.167 | | | | |
| **MATERNAL FACTORS** | | | | | | | | | |
| **Age (years)** | **21-29 vs. ≤ 20** | 0.981 | 0.916 | 0.682 | 1.409 | | | | |
| | **≥30 vs. ≤ 20** | 0.796 | 0.296 | 0.520 | 1.221 | | | | |
| **Ethnicity** | **Non-Afro vs. Afro** | **0.460** | **<0.001** | **0.332** | **0.639** | **0.640** | **0.006** | **0.465** | **0.881** |
| **Educational status** | **Primary vs. illiterate** | **0.657** | **0.073** | **0.416** | **1.040** | | | | |
| | **Secondary vs. illiterate** | **0.416** | **0.001** | **0.250** | **0.694** | | | | |
| **HOUSEHOLD FACTORS** | | | | | | | | | |
| **Socio-economic status** | **Medium vs. Low** | 0.768 | 0.158 | 0.532 | 1.108 | | | | |
| | **High vs. Low** | 0.808 | 0.258 | 0.558 | 1.169 | | | | |
| **Area of residence** | **Rural vs. Urban** | **0.715** | **0.042** | **0.518** | **0.989** | **0.733** | **0.048** | **0.539** | **0.998** |
| **Overcrowding** | **≥3 vs. < 3** | **1.695** | **0.001** | **1.250** | **2.298** | | | | |
| **Monthly income (basic basket)** | **≥1 vs. < 1** | **0.368** | **0.012** | **0.169** | **0.802** | **0.435** | **0.023** | **0.212** | **0.891** |
| **House construction** | **Cement-brick vs. Wood/bamboo** | 0.867 | 0.404 | 0.621 | 1.212 | | | | |
| **Dog in house (birth)** | **Yes vs. No** | 1.059 | 0.811 | 0.664 | 1.688 | | | | |
| **Cat in house (birth)** | **Yes vs. No** | 1.318 | 0.232 | 0.838 | 2.072 | | | | |
| **Agriculture (TV)** | **Yes vs. No** | 1.114 | 0.488 | 0.821 | 1.512 | | | | |
| **Bathroom (TV)** | **WC vs. latrine** | 0.851 | 0.280 | 0.634 | 1.141 | | | | |
| **HOUSEHOLD STH INFECTIONS** | | | | | | | | | |
| **Mother** | **Yes vs. No** | **2.518** | **<0.001** | **1.876** | **3.379** | **1.985** | **<0.001** | **1.484** | **2.657** |
| **Father** | **Yes vs. No** | **2.218** | **0.003** | **1.313** | **3.746** | | | | |
| **Siblings** | **Yes vs. No** | **1.972** | **0.001** | **1.299** | **2.994** | | | | |
| **Any in household** | **Yes vs. No** | **1.794** | **0.001** | **1.282** | **2.510** | | | | |

Odds ratios (ORs) and 95% confidence intervals (CI) were estimated by fitting (age, age$^2$, age$^3$, age$^4$, age$^5$, last two estimates not shown) adjusted ordinal longitudinal models using generalized estimating equations. The ORs represents the effects of each variable on the odds of having >=2, >=1 infections vs the lower category, namely <2, < 1, respectively. Longitudinal ordinal logistic models were fit under missing completely at random assumption for unobserved data points. Characteristics are at time of birth of child (birth) or time-varying (TV) over the course of follow-up. STH—soil-transmitted helminth infections. Afro- Afro-Ecuadorian. Anthelmintic treatments—maternal reports of at least one anthelmintic treatment during the previous year. Monthly household income was classified according to receipt of an income sufficient to meet the basic needs of 4 persons (or 1 family basket) or US$480 in 2008. Household overcrowding was defined as 3 or more people per sleeping room. Agricultural exposures were defined by living on a farm or having at least weekly visits to a farm. Any infected with STH in the household represented any member of the child's household with a positive stool sample collected around the time of birth of the cohort child.

**Table 3. Multivariable associations of any parasite infection (IPI), any soil-transmitted helminth infection (STH), or any protozoa infection with longitudinal weight-for-age and height-for age z scores.**

| Variable | Categories | Weight-for-age (z-score) | | | | Height-for-age (z-score) | | | |
|---|---|---|---|---|---|---|---|---|---|
| | | Estimate | P-value | 95% CI low | 95% CI high | Estimate | P-value | 95% CI low | 95% CI high |
| **Any IPI** | **YES vs. NO** | **-0.129** | **0.018** | **-0.237** | **-0.022** | **-0.126** | **0.021** | **-0.233** | **-0.019** |
| **Any STH** | **YES vs. NO** | **-0.212** | **0.001** | **-0.339** | **-0.085** | **-0.180** | **0.006** | **-0.307** | **-0.053** |
| **Any protozoa** | **YES vs. NO** | -0.065 | 0.053 | -1.230 | 0.220 | **-0.090** | **0.012** | **-0.193** | **0.012** |
| **Multiparasitism** | **1 vs. 0** | -0.099 | 0.091 | -0.214 | 0.016 | -0.073 | 0.219 | -0.189 | 0.043 |
| | **≥2 vs. 0** | **-0.228** | **0.003** | **-0.379** | **-0.077** | **-0.289** | **<0.001** | **-0.441** | **-0.137** |
| | **≥2 vs. 1** | -0.128 | 0.094 | -0.279 | -0.022 | **-0.216** | **0.005** | **-0.368** | **-0.064** |

The estimates and their 95% confidence intervals (CI) are adjusted for age, age$^2$, age$^3$, and age$^4$ and childhood, maternal, and household factors as explained in Methods using generalized estimating equations for normally distributed longitudinal growth outcomes. Models fit under missing completely at random assumption for unobserved data points. Estimates represent the adjusted average differences in z-scores between positive and negative children for IPI, any protozoa, any STH, and multiparasitism evaluated at the same time.

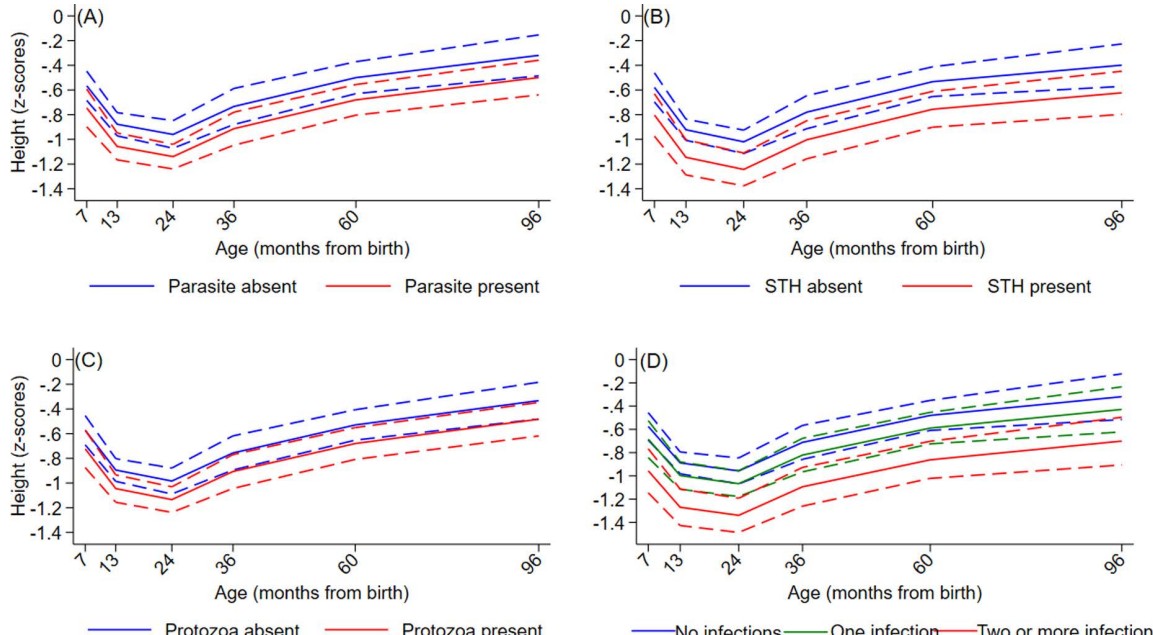

**Fig 3. Effects of *intestinal parasite infections* on trajectories of height-for-age z scores during childhood up to 8 years.** A – Any intestinal parasite (Parasite) infection. B- any soil-transmitted helminth (STH) parasite. C – any intestinal protozoal (Protozoa) parasite. D – Multiparasitism. Y axes show z scores. Interrupted curves represent 95% confidence intervals.

8 years of life compared to the group without infection (Figs 3A, 3B, 4A and 4B). For any protozoa infection, there was a significant effect on HAZ (Fig 3C) while the effect on WAZ was borderline significant (Fig 4C). Similarly, significantly lower HAZ and WAZ trajectories were seen for the group of children with multiparasitism (>=2 infections vs. 0 infections) with evidence of greater reductions in HAZ and WAZ with greater number of parasite species (Figs 3D and 4D), particularly for HAZ. Significant rates of stunting (HAZ z scores <=-2) were present in the cohort reaching 15.2% at 24 months of age (S1A Fig) - there was some evidence of increased stunting among children with STH infections (S1C Fig) or with multiparasitism (S1D Fig), although these were not statistically significant. Sensitivity analyses for missing values for parasite data for estimates of effect on WAZ and HAZ showed very similar effects between models.

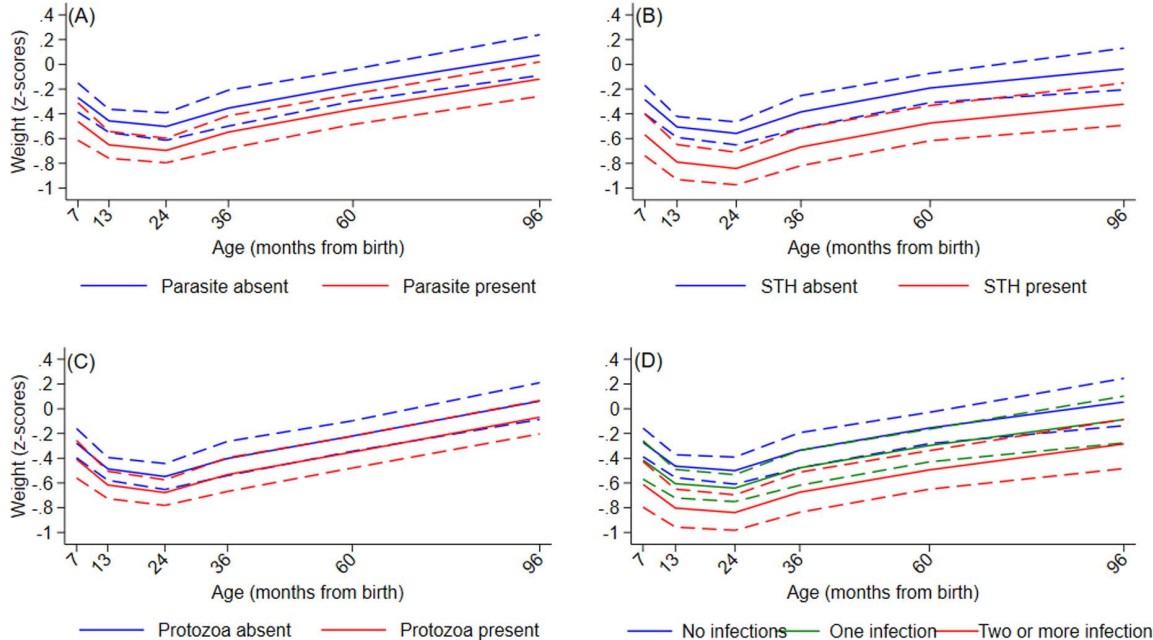

**Fig 4. Effects of intestinal parasite infections on trajectories of weight-for-age z scores during childhood up to 8 years.** A – Any intestinal parasite (Parasite) infection. B- any soil-transmitted helminth (STH) parasite. C – any intestinal protozoal (Protozoa) parasite. D – Multiparasitism. Y axes show z scores. Interrupted curves represent 95% confidence intervals.

### Effects of IPI and multiparasite infections on childhood anemia and hemoglobin levels

Age-adjusted and multivariable associations between infections with any IPI, any STH, and any protozoa and anemia or hemoglobin levels in Tables 4 and S5, respectively. In multivariable analyses, we did not observe significant effects of parasite infections on the risk of anemia. However, any IPI was associated with significantly reduced hemoglobin over follow up (S2A Fig). When we considered previous infections as potential determinants of current hemoglobin levels, any STH was significantly associated with reduced hemoglobin trajectories (Table 4). Sensitivity analyses for missing values for parasite data for estimates of effect on hemoglobin showed very similar effects between models.

### Effects of IPI and multiparasite infections on childhood eosinophilia

Age-adjusted and multivariable associations between any IPI, any protozoa, and any STH and childhood trajectories of eosinophilia are shown in S6 and S7 Tables, respectively. In multivariable analyses, current infections with any IPI (S2A Fig), any STH (S2B Fig), and any protozoa (S2C Fig) were significantly associated with trajectories for presence of eosinophilia (S7 Table). Interestingly, previous infections with any STH were also significantly associated with (current) eosinophilia, while there was no evidence to suggest a similar effect for previous IPI or protozoa.

## Discussion

In the present study, we explored determinants of intestinal parasite infections and multiparasitism (or polyparasitism) and the effects of these infections during childhood on growth and nutrition. To do this we used data from a random sample of children nested within a population-based birth cohort that was recruited in a rural coastal district in Ecuador. We used highly sensitive molecular methods (multi-parallel quantitative real-time PCR) to detect infections with intestinal parasites including those soil-transmitted helminths and protozoa that have been shown to be endemic in this setting previously

**Table 4. Multivariable associations of any parasite infection (IPI), any soil-transmitted helminth infection, or any protozoa infection with longitudinal risk of anaemia (defined as <11 g/dL) and hemoglobin levels (g/dL) between 7 months and 8 years of age. ORs/estimates measure adjusted associations between presence/absence of infection when evaluated at the same time as anemia/hemoglobin (current infections) or at the previous observation time (previous infections) during follow-up.**

| Parasite infection | Categories | Anemia | | | | Hemoglobin | | | |
|---|---|---|---|---|---|---|---|---|---|
| | | OR | P-value | 95% CI low | 95% CI high | Estimate | P-value | 95% CI low | 95% CI high |
| **Current infections** | | | | | | | | | |
| **Any IPI** | Yes vs. No | 1.205 | 0.265 | 0.868 | 1.676 | **-0.102** | **0.025** | **-0.192** | **-0.013** |
| **Any STH** | Yes vs. No | 1.024 | 0.919 | 0.646 | 1.623 | -0.088 | 0.110 | -.0196 | 0.020 |
| **Any protozoa** | Yes vs. No | 1.196 | 0.289 | 0.860 | 1.663 | -0.066 | 0.133 | -0.153 | 0.020 |
| **Multiparasitism** | 1 vs. 0 | 1.273 | 0.178 | 0.896 | 1.811 | **-0.126** | **0.015** | **-0.226** | **-0.025** |
| | ≥2 vs. 0 | 1.044 | 0.876 | 0.606 | 1.800 | -0.053 | 0.417 | -0.182 | 0.076 |
| | ≥2 vs. 1 | 0.820 | 0.480 | 0.472 | 1.424 | 0.072 | 0.281 | -0.059 | 0.203 |
| **Previous infections** | | | | | | | | | |
| **Any IPI** | Yes vs. No | 1.093 | 0.692 | 0.704 | 1.698 | -0.093 | 0.069 | -0.194 | 0.007 |
| **Any STH** | Yes vs. No | 1.364 | 0.310 | 0.749 | 2.485 | **-0.134** | **0.034** | **-0.259** | **-0.010** |
| **Any protozoa** | Yes vs. No | 1.061 | 0.795 | 0.679 | 1.658 | -0.098 | 0.056 | -0.198 | 0.002 |

Estimates show adjusted effects of current (measured at same time as hemoglobin) and represent the adjusted average difference in haemoglobin levels between positive and negative (for infections) children evaluated at the time of haemoglobin measurement or on the previous occasion. Estimates (ORs) and 95% confidence intervals (CI) for anemia were estimated by fitting age and age$^2$-adjusted models using generalized estimating equations tailored to binary longitudinal data. Models were adjusted for childhood, maternal, and household factors as explained in Methods, and were fit under missing completely at random assumption for unobserved data points.

[9,16,29]. Our data showed a high prevalence with both STH and intestinal protozoal parasites after the first year of life and the gradual acquisition of multiparasitism. IPI and multiparasitism were more frequent among children of Afro-Ecuadorian ethnicity living in less affluent and STH-infected households, and among those attending day care, all indicating the role of poverty and exposure to contaminated environments in determining infection risk during childhood. Childhood IPI were associated with reduced height and weight trajectories during the first 8 years of life, particularly among those children with multiparasitism. We did not detect effects of IPI on childhood anemia but did observe lower hemoglobin levels among infected children. Overall, our data show that IPI and multiparasitism are frequent during childhood in an endemic region of coastal Ecuador and that these infections are associated with reduced growth trajectories and impaired nutrition.

Although a quarter of the world population has been estimated to have intestinal parasites [5–7] frequently with multiple parasite species, there are limited data from longitudinal studies on determinants of multiparasitism and their effects on growth and nutrition in children. Among IPI, STH infections are rarely fatal but have been associated with chronic undernutrition causing impaired growth and iron-deficiency anemia [3,30]. The most widely used indicator for chronic malnutrition is the presence of stunting, representing a failure in linear growth, caused by a lack of access to and/or absorption of sufficient quantities of specific nutrients during childhood [31,32]. Linear growth is largely determined between 6 and 24 months of age [33], the age range most affected any mismatch between nutritional demands and nutrient supply [34,35]: in this study, average HAZ z scores were lowest at 2 years, and the group most affected by low HAZ z scores were children with multiparasitism.

IPI may affect childhood growth and nutrition that may be caused by anorexia, intestinal bleeding, malabsorption, chronic diarrhea, and competition for nutrients [36–38]. Exposures to IPI and other enteric pathogens [39] may cause chronic activation of gut immune cells, resulting in environmental enteric dysfunction (EED) [35]. EED is characterized by gut mucosal cell villous atrophy, crypt hyperplasia, increased permeability and inflammatory cell infiltration, and microbial dysbiosis [35]. Chronic EED is considered to contribute to chronic undernutrition and in combination with repeated infections causes nutrient malabsorption resulting in wasting, stunting, and anemia [35]. Among IPI, *A. lumbricoides* and *G.*

*lamblia* are considered important causes of undernutrition in under-5s [30,40], while *T. trichiura* may contribute to iron-deficiency anemia through its direct effects on mucosal bleeding in the large intestine [30]. Further, multiparasitism and higher parasite burdens with individual parasites likely increase the risk of clinically relevant undernutrition.

Previous studies have shown IPI to be associated more strongly with stunting than wasting (i.e., weight-for-age z score <=2 SD) reflecting the importance of the long-term nutritional effects of chronic parasitism [40]. Similarly, in this study we observed greater effects of IPI on HAZ compared to than WAZ z scores. A previous systematic review estimated that IPIs were associated with 2.5-times greater odds of stunting, particularly infections through with *A. lumbricoides*, *E. histolytica* and, *G. lamblia* [40]. Evidence for a role of STH in causing stunting remains controversial [41]. Certainly, individual observational studies have reported reduced linear growth being associated with infections with *A. lumbricoides* and *T. trichiura* [42,43], particularly moderate to heavy infection burdens [14,44,45] or co-infections with both parasites [46], but the potential for confounding by unmeasured causal factors remains a problem. A recent systematic review of randomized trials and observational studies did not provide strong evidence for improved growth among children receiving anthelmintic treatment [41]. However, the findings of systematic reviews [41,47] have been criticized because of the age groups studied (i.e., only 2 published studies of under-5s [48,49]), relatively short follow-up period to observe chronic nutritional effects (e.g., follow-up periods generally 1 year or less are unlikely to show effects on linear growth), and the dilutional effects of treating both infected and uninfected children. Our study, that unusually covered not only the critical period for linear growth (i.e., 6 to 24 months) but also the whole of early childhood to school age, therefore, provides novel evidence for an effect of IPI on linear growth although the relatively small sample size did not allow us to determine parasite species-specific effects.

Asymptomatic *G. lamblia* infection has been associated with reduced linear growth in under-5s previously [29,50,51]. We did observe stronger effects of STH than protozoal infections on linear growth deficits although the strong association between the STH and protozoal parasites made it difficult to distinguish independent effects. Impaired linear growth and stunting is associated with reduced physical and cognitive growth [52] with long-term effects not only on individuals through lower levels of schooling and household income [53], but also societies through decreased national economic output [52,54].

We measured hemoglobin levels and anemia as an indicator for micronutrient deficiency - in this case iron deficiency. We were unable to detect effects of IPI on childhood anemia but did observe lower hemoglobin levels among infected children but with no evidence of a greater effect of multiparasitism compared to monoparasitism. Interestingly, there was some evidence that previous STH but not protozoal infections might predict hemoglobin levels over the course of follow-up. STH may reduce hemoglobin levels either through direct blood loss from the intestinal surface (e.g., hookworm and *T. trichiura*) [3,30], through impaired iron absorption in the context of a deficient diet (perhaps, *A. lumbricoides* [55]), or through effects in promoting mucosal and systemic inflammation (many IPI and STH pathogens). Meta-analyses of randomized and quasi-randomized trials of anthelmintic drugs have failed to show significant effects of STH parasites on rates of anemia [47,56]. However, there was some evidence for an effect on anemia from studies done in Sub-Saharan Africa where hookworm prevalence is greater [56]. Hookworm prevalence was low in this birth cohort but our analysis of the effect of STH on hemoglobin levels likely allowed us to detect more subtle effects of these parasites on host morbidity.

Numerous epidemiological studies have estimated IPI prevalence in pre-school and school age children in different populations. A recent meta-analysis of 83 studies of IPI done between 1996 and 2019 in pre-school and school age children in Ethiopia estimated a prevalence of 48% with 16% of infections being multiparasitic [57]. A meta-analysis of 32 studies of IPI prevalence in Brazilian children between 2000 and 2018 estimated an overall prevalence of 46% with the greatest prevalence (58%) in the country's poorest North region with no specific data being provided for rates of multiparasitism [58]. Almost all these studies used microscopic detection methods likely to underestimate prevalence compared to more sensitive molecular methods. A previous study of children aged 3 to 11 years in rural communities in the Provinces

**PLOS** Neglected Tropical Diseases

of Chimborazo and Guayas in Ecuador estimated a prevalence of 63.2% by microscopy [59]. In this longitudinal study, we detected maximal prevalence of IPI of 69.5% at 8 years age at which age rates of multiparasitism were 32.5%.

Important risk factors for IPI from previous studies include rural residence, low household income, having illiterate parents, household overcrowding, and lack of access to potable water and sanitation [40,59]. Here, risk factors for IPI included being lower in the birth order (i.e., having more older siblings), Afro-Ecuadorian ethnicity (here indicating a marginalized population group), and having an STH-infected mother (likely indicator for infection risk within the household). Some risk factors for STH and protozoal infections were shared (i.e., vaginal delivery and maternal STH), while other were distinct for STH (i.e., Afro-Ecuadorian ethnicity, overcrowding, and agricultural exposures) and protozoal infections (male sex, lower in birth order, day care, and urban residence). In this setting, urban residence generally reflected a peri-urban household built in informal settlements at the growing periphery of a town with limited access to basic services. Sanitation was not specifically associated with IPI risk, possibly because of a lack of heterogeneity in this exposure in our study population representing the poorest segment of the population dependent on public provision for health care.

## Strengths and limitations

There are few longitudinal studies of IPI in infants and pre-school children, the groups most vulnerable to nutritional effects of enteric parasites, from any region. We were able to evaluate the effects of these infections, both individually and as multiparasitic infections, on growth and nutritional indices during the first 8 years of life. Further, we used molecular detection methods sufficiently sensitive to detect light infections that are common in infants and pre-school children [60]. The study of a population in a setting where systemic helminth (e.g., trematode and filarial worms) and protozoal pathogens (e.g., malaria) - that may themselves modify childhood growth and nutrition - are largely absent, allowed us to more clearly evaluate the effects of enteric polyparasitism on study outcomes. The collection of data on a wide variety of relevant individual, maternal, and household factors including an indicator for IPI infection risk (i.e., presence of STH infections in household members around the time of the child's birth) allowed us to control for confounding although we cannot exclude residual confounding by unmeasured factors. Potential unmeasured confounders include micronutrient supplementation and dietary quality both of which could affect nutritional outcomes, but which have less clear effects on longitudinal trajectories of IPI risk. The prevalence of stunting in this birth cohort sub-sample was probably too low to allow us to show significant effects of IPI or parasite sub-groups on this outcome. The lower rate of stunting compared to national estimates in Ecuador in 2014 of 25.3% [61] is likely explained by improved health in cohort members through regular contacts with health professionals in the study clinic – benefits included monitoring of nutritional supplement intake (provided free by the public health system) and anti-parasite treatments for positive stool samples. Frequent antiparasitic treatments will have reduced prevalence and intensity of IPI and our ability to detect effects on growth. As for stunting, the prevalence of anemia was probably too low to detect significant effects of IPI on this outcome - the low prevalence of hookworm, a major cause of anemia in STH-endemic populations, would have further limited anemia risk. We separated IPI into those caused by STH and protozoal parasites, but again because of relatively small numbers were unable to explore parasite-parasite interactions, effects of specific parasites, or effects of infection burdens with individual parasites. The dominant STH infections in this setting were *A. lumbricoides* and *T. trichiura* while the dominant protozoal parasite was *G. lamblia* – observed associations for or effects by STH and protozoal parasites can therefore be explained largely by these 3 parasites.

## Conclusion

We followed a random sample from a birth cohort to 8 years in an area of Ecuador endemic for IPI and monitored the acquisition of IPI and multiparasitism during childhood using multi-parallel quantitative real-time PCR, and evaluated their effects on growth and nutritional outcomes. IPI and multiparasitism during childhood were associated with indicators of

poverty and household risk of STH. IPI was associated with lower HAZ and WAZ and hemoglobin levels during childhood. There was evidence for stronger effects on growth outcomes of multiparasitic compared to monoparasitic or no infections. Previous STH infections appeared to more strongly predict lower hemoglobin levels and the presence of eosinophilia than did protozoal infections. Our data provide evidence of a high endemicity of IPI during childhood in this tropical setting, and of their adverse effects on growth and nutrition during childhood, effects that were generally more pronounced in children with multiparasitic infections. This study helps fill an unmet need for data on the health impact of IPI and multiparasitism in under-5s. Future larger studies could explore effects on growth and nutrition of individual parasite species and interactions between them, as well as the impact of regular antiparasitic treatments with broad spectrum drugs (e.g., nitazoxanide) with anthelmintic and antiprotozoal effects, on growth and nutritional outcomes during childhood. Improved mapping of high-endemicity communities for IPI, where targeted or community-wide antiparasitic treatments could be focused, would enhance the impact of this strategy.

## Supporting information

**S1 Table. Descriptive statistics for frequences of infections with any STH, any protozoa, and any parasite in 401 children according to categories of childhood, maternal, and household factors.** Summaries are made upon ever/never infected to preserve the independence of the observations. Infection frequencies represent any infection identified during follow-up from 7 months to 8 years of age. Frequencies for categories of time-varying variables represent data recorded at birth. Characteristics are at time of birth of child or time-varying (TV) over the course of follow-up. STH—soil-transmitted helminth infections. Afro- Afro-Ecuadorian. Anthelmintic treatments—maternal reports of at least one anthelmintic treatment during the previous year. Monthly household income was classified according to receipt of an income sufficient to meet the basic needs of 4 persons (or 1 family basket) or US$480 in 2008. Household overcrowding was defined as 3 or more people per sleeping room. Agricultural exposures were defined by living on a farm or having at least weekly visits to a farm. Any infected with STH in the household represented any member of the child's household with a positive stool sample collected around the time of birth of the cohort child.
(DOCX)

**S2 Table. Age-adjusted associations between child, maternal, and household factors with infections with any soil-transmitted helminth (STH) or protozoal parasite from 7 months to 8 years of age.** Odds ratios (ORs) and 95% confidence intervals (CI) were estimated by fitting age, age2, and age3-adjusted longitudinal models using generalized estimating equations. Longitudinal binary outcomes were defined by presence/absence of any STH or any protozoa detected in stool samples from children during follow-up. Models were fit under missing completely at random assumption for unobserved data points. Characteristics are at time of birth of child (birth) or time-varying (TV) over the course of follow-up. STH—soil-transmitted helminth infections. Afro- Afro-Ecuadorian. Anthelmintic treatments—maternal reports of at least one anthelmintic treatment during the previous year. Monthly household income was classified according to receipt of an income sufficient to meet the basic needs of 4 persons (or 1 family basket) or US$480 in 2008. Household overcrowding was defined as 3 or more people per sleeping room. Agricultural exposures were defined by living on a farm or having at least weekly visits to a farm. Any infected with STH in the household represented any member of the child's household with a positive stool sample collected around the time of birth of the cohort child.
(DOCX)

**S3 Table. Multivariable associations between child, maternal, and household factors with infections with any soil-transmitted helminth (STH) or protozoal parasite from 7 months to 8 years of age.** Odds ratios (ORs) and 95% confidence intervals (CI) were estimated by fitting age, age2, and age3-adjusted longitudinal models using generalized estimating equations. Longitudinal binary outcomes were defined by presence/absence of any STH or any protozoa detected

in stool samples from children during follow-up. Models were fit under missing completely at random assumption for unobserved data points. Characteristics are at time of birth of child (birth) or time-varying (TV) over the course of follow-up. STH—soil-transmitted helminth infections. Afro- Afro-Ecuadorian. Household overcrowding was defined as 3 or more people per sleeping room. Agricultural exposures were defined by living on a farm or having at least weekly visits to a farm.
(DOCX)

**S4 Table. Age-adjusted associations of any parasite infection (IPI), any soil-transmitted helminth infection (STH), or any protozoa infection with longitudinal weight-for-age and height-for age z scores (g/dL).** Estimates are adjusted for weight or length at birth, respectively. Estimates (ORs) and 95% confidence intervals (CI) were estimated by fitting age, age2, age3, and age4-adjusted longitudinal models using generalized estimating equations. Models were fit under missing completely at random assumption for unobserved data points.
(DOCX)

**S5 Table. Age-adjusted associations of any parasite infection (IPI), any soil-transmitted helminth infection, or any protozoa infection with longitudinal risk of anemia and hemoglobin levels (g/dL) between 7 months and 8 years of age.** Estimates show effects of current (measured at same time as hemoglobin) and previous (measured at previous time point) infections on longitudinal risk of anemia and average levels of hemoglobin. Estimates (ORs) and 95% confidence intervals (CI) were estimated by fitting age and age2-adjusted longitudinal models using generalized estimating equations. Models were fit under missing completely at random assumption for unobserved data points.
(DOCX)

**S6 Table. Age-adjusted associations of any parasite infection (IPI), any soil-transmitted helminth infection, or any protozoa infection with longitudinal risk of eosinophilia between 7 months and 8 years of age.** Estimates show effects of current (measured at same time as eosinophilia) and previous (measured at previous time point) infections on longitudinal risk of eosinophilia.
(DOCX)

**S7 Table. Multivariable associations of any parasite infection (IPI), any soil-transmitted helminth infection, or any protozoa infection with longitudinal risk of eosinophilia between 7 months and 8 years of age.** Estimates show effects of current (measured at same time as eosinophilia) and previous (measured at previous time point) infections on longitudinal risk of eosinophilia.
(DOCX)

**S1 Fig. Effects of intestinal parasite infections on proportions of children with stunting during childhood up to 8 years.** A – All children. B - Any soil-transmitted helminth (STH) parasite. C – Any intestinal parasite (Parasite) infection. D - Any intestinal protozoal (Protozoa) parasite. E – Multiparasitism. Y axes show proportions with stunting (defined as height-for-age z scores <=2). Interrupted curves represent 95% confidence intervals.
(TIF)

**S2 Fig. Effects of intestinal parasite infections on trajectories of hemoglobin (g/dL) during childhood up to 8 years.** A – Any intestinal parasite (Parasite) infection. B- any soil-transmitted helminth (STH) parasite. C – any intestinal protozoal (Protozoa) parasite. D – Multiparasitism. Y axes show hemoglobin (g/dL). Interrupted curves represent 95% confidence intervals.
(TIF)

**S3 Fig. Effects of intestinal parasite infections on proportions of children with eosinophilia during childhood up to 8 years.** A – Any intestinal parasite (Parasite) infection. B- any soil-transmitted helminth (STH) parasite. C – any

intestinal protozoal (Protozoa) parasite. D – Multiparasitism. Y axes show proportions with eosinophilia (defined as>= 500 cells/µL). Interrupted curves represent 95% confidence intervals.
(TIF)

**S1 Data. Raw data used for analyses.**
(XLSX)

## Acknowledgments

We thank the ECUAVIDA study team for their dedicated work and the cohort mothers and children for their enthusiastic participation. We acknowledge the support of the Ecuadorian Ministry of Public Health and the Directors and Staff of the Hospital "Padre Alberto Buffoni", Quinindé, Esmeraldas Province.

## Author contributions

**Conceptualization:** Rojelio Mejia, Philip J. Cooper.

**Data curation:** Martha E Chico.

**Formal analysis:** Irina Chis Ster.

**Funding acquisition:** Rojelio Mejia, Philip J. Cooper.

**Investigation:** Rojelio Mejia, Martha E Chico, Irene Guadalupe, Andrea Arévalo-Cortés, Andrea Lopez, Aida Y Oviedo-Vera, Philip J. Cooper.

**Methodology:** Rojelio Mejia, Philip J. Cooper.

**Project administration:** Martha E Chico, Philip J. Cooper.

**Resources:** Philip J. Cooper.

**Supervision:** Martha E Chico, Philip J. Cooper.

**Visualization:** Irina Chis Ster.

**Writing – original draft:** Irina Chis Ster, Philip J. Cooper.

**Writing – review & editing:** Rojelio Mejia, Martha E Chico, Irene Guadalupe, Andrea Arévalo-Cortés, Andrea Lopez, Aida Y Oviedo-Vera.

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
