## [Decision Letter · Decision Letter 0]

24 Apr 2025

Response to Reviewers
Revised Manuscript with Track Changes
Manuscript

co-Editor-in-Chief

Paul Brindley

co-Editor-in-Chief

**Additional Editor Comments :**
**Journal Requirements:**

At this stage, the following Authors/Authors require contributions: Rojelio Mejia, Irina Chis Ster, Martha E Chico, Irene Guadalupe, Andrea Arevalo-Cortes, Andrea Lopez, Aida Oviedo, and Philip J. Cooper. Please ensure that the full contributions of each author are acknowledged in the "Add/Edit/Remove Authors" section of our submission form.

2) We noticed that you used the phrase 'data not shown' in the manuscript. We do not allow these references, as the PLOS data access policy requires that all data be either published with the manuscript or made available in a publicly accessible database. Please amend the supplementary material to include the referenced data or remove the references.

**Reviewers' comments:**

**Key Review Criteria Required for Acceptance?**

**Methods**

-Are the objectives of the study clearly articulated with a clear testable hypothesis stated?

-Is the study design appropriate to address the stated objectives?

-Is the population clearly described and appropriate for the hypothesis being tested?

-Is the sample size sufficient to ensure adequate power to address the hypothesis being tested?

-Were correct statistical analysis used to support conclusions?

-Are there concerns about ethical or regulatory requirements being met?

Reviewer #1: (No Response)

Reviewer #2: Yes to all the above

**Results**

-Does the analysis presented match the analysis plan?

-Are the results clearly and completely presented?

-Are the figures (Tables, Images) of sufficient quality for clarity?

Reviewer #1: (No Response)

Reviewer #2: Yes to all the above

**Conclusions**

-Are the conclusions supported by the data presented?

-Are the limitations of analysis clearly described?

-Do the authors discuss how these data can be helpful to advance our understanding of the topic under study?

-Is public health relevance addressed?

Reviewer #1: (No Response)

Reviewer #2: YES to all the above

**Editorial and Data Presentation Modifications?**

Reviewer #1: (No Response)

Reviewer #2: Minor corrections

Abstract:

Methods - rephrase "studies for 8 years"

Results - Greater parasite numbers suggests intensity to me perhaps rephrase for clarity

Methods:

Maternal questionnaires were repeated when the child was... p. 8

Rephrase : Fit back" p. 11

Discussion:

Other systemic helminths were absent - please clarify p. 27

**Summary and General Comments**

Reviewer #1: The manuscript presents a comprehensive longitudinal study on the epidemiology of intestinal parasite infections (IPIs) and multiparasitism in a birth cohort from tropical Ecuador, utilizing molecular detection methods. The study addresses an important gap in understanding the impact of IPIs on childhood growth and hemoglobin levels. The findings are relevant to public health, particularly in endemic regions, and align with WHO priorities for reducing morbidity associated with parasitic infections. The study design is robust, and the use of molecular methods enhances the sensitivity of parasite detection. However, some areas require clarification or improvement to strengthen the manuscript.

1. Please use updated data and valid references for statistics on IPI-related prevalence and DALYs presented in the introduction.

2. Please clarify whether qPCR testing was performed on all samples at each time point or only on those that were microscopy-positive.

3. Please clarify how other unmeasured potential confounders (e.g., dietary intake, micronutrient supplementation) that could influence growth and hemoglobin outcomes might affect the results of this study.

4. The lack of a significant effect on anemia (despite lower hemoglobin levels) warrants further discussion. Could this be due to the low prevalence of hookworm (a major driver of anemia) in the cohort?

Reviewer #2: The authors should be commended on this powerful study with meticulous detail throughout. In particular, the major strength of the study is its longitudinal design covering child age from early childhood to school age. Consequently the impact of parasitism is clearly shown in contrast to short-term follow-up studies that mainly informed recent meta-analyses. Furthermore, the use of sensitive molecular detection methods allowed the identification of low intensity infections. As pointed out by the authors, the impact of specific parasite species was not possible but this study highlights the methodology in which future studies can be designed to delineate such species-specific impacts.

PLOS authors have the option to publish the peer review history of their article (what does this mean? ). If published, this will include your full peer review and any attached files.

**Do you want your identity to be public for this peer review?** For information about this choice, including consent withdrawal, please see our Privacy Policy .

Reviewer #1: No

Reviewer #2: No

**Figure resubmission:****Reproducibility:** To enhance the reproducibility of your results, we recommend that authors of applicable studies deposit laboratory protocols in protocols.io, where a protocol can be assigned its own identifier (DOI) such that it can be cited independently in the future. Additionally, PLOS ONE offers an option to publish peer-reviewed clinical study protocols. Read more information on sharing protocols at https://plos.org/protocols?utm_medium=editorial-email&utm_source=authorletters&utm_campaign=protocols

---

## [Editor Report · Decision Letter 1]

26 May 2025

Dear Dr. Cooper,

We are pleased to inform you that your manuscript 'Epidemiology of intestinal parasite infections and multiparasitism and their impact on growth and hemoglobin levels during childhood in tropical Ecuador: a longitudinal study using molecular detection methods.' has been provisionally accepted for publication in PLOS Neglected Tropical Diseases.

Best regards,

jong-Yil Chai

Section Editor

Jong-Yil Chai

Section Editor

Shaden Kamhawi

co-Editor-in-Chief

Paul Brindley

co-Editor-in-Chief

The revised manuscript is now acceptable for publication in PLoS NTD.

---

## [Editor Report · Acceptance letter]

Dear Dr. Cooper,

We are delighted to inform you that your manuscript, "Epidemiology of intestinal parasite infections and multiparasitism and their impact on growth and hemoglobin levels during childhood in tropical Ecuador: a longitudinal study using molecular detection methods.," has been formally accepted for publication in PLOS Neglected Tropical Diseases.

Best regards,

Shaden Kamhawi

co-Editor-in-Chief

Paul Brindley

co-Editor-in-Chief
